# Profiling of Tumour-Infiltrating Lymphocytes and Tumour-Associated Macrophages in Ovarian Epithelial Cancer—Relation to Tumour Characteristics and Impact on Prognosis

**DOI:** 10.3390/ijms25084524

**Published:** 2024-04-20

**Authors:** Annabel Stout, Natalya Facey, Anjali Bhatnagar, Kirstie Rice, Fedor Berditchevski, Daniel Kearns, Amy Metcalf, Alaa Elghobashy, Abeer M. Shaaban

**Affiliations:** 1Department of Gynaecological Oncology, Birmingham Women’s Hospital, Edgbaston, Birmingham B15 2TG, UK; annabel.stout@nhs.net; 2Department of Cellular Pathology, University Hospitals Birmingham NHS Foundation Trust, Queen Elizabeth Hospital Birmingham, Edgbaston, Birmingham B15 2GW, UK; faceynatalya@yahoo.co.uk (N.F.); daniel.kearns@uhb.nhs.uk (D.K.); 3Department of Cellular Pathology, The Royal Wolverhampton NHS Trust, New Cross Hospital, Wolverhampton WV10 0QP, UK; anjali.bhatnagar@nhs.net (A.B.); kirstie.rice@nhs.net (K.R.); 4Institute of Cancer and Genomic Sciences, University of Birmingham, Edgbaston, Birmingham B15 2TT, UK; f.berditchevski@bham.ac.uk; 5Department of Cellular Pathology, University Hospitals Birmingham NHS Foundation Trust, Birmingham Heartlands Hospital, Bordesley Green East, Birmingham B9 5SS, UK; amy.metcalf@uhb.nhs.uk; 6Department of Gynaecological Oncology, The Royal Wolverhampton NHS Trust, New Cross Hospital, Wolverhampton WV10 0QP, UK; alaaelghobashy@nhs.net

**Keywords:** ovarian cancer, CD3, CD20, CD163, CD68, tumour-infiltrating lymphocyte, TIL

## Abstract

Early evidence suggests a strong impact of tumour-infiltrating lymphocytes (TILs) on both the prognosis and clinical behaviour of ovarian cancer. Proven associations, however, have not yet translated to successful immunotherapies and further work in the field is urgently needed. We aimed to analyse the tumour microenvironment of a well-characterised cohort of ovarian cancer samples. Tumour markers were selected owing to their comparative underrepresentation in the current literature. Paraffin-embedded, formalin-fixed tumour tissue blocks of 138 patients representative of the population and including early stage disease were identified, stained for CD3, CD20, CD68 and CD163 and analysed for both the stromal and intertumoral components. Data were statistically analysed in relation to clinical details, histological subtype, borderline vs. malignant status, survival and management received. Mean stromal CD3, total CD3 count, mean stromal CD20 and total CD20 count all correlated negatively with survival. Malignant ovarian tumours consistently demonstrated significantly higher infiltration of all analysed immune cells than borderline tumours. Assessment of the stromal compartment produced a considerably higher proportion of significant results when compared to the intra-tumoural infiltrates. Customary assessment of solely intra-tumoural cells in advanced stage disease patients undergoing primary debulking surgery should be challenged, with recommendations for future scoring systems provided.

## 1. Introduction

Ovarian cancer carries the worst prognosis of any gynaecological malignancy. Despite successful initial management, high recurrence rates of 70–85% lead to a 10-year survival rate of only 35% [1]. The interaction between epithelial ovarian cancer cells and the immune system has long been described; however, progression in the understanding of underlying mechanisms and their impact on management has fallen behind other primary malignancies [2]. Advances have been seen in the immune modulation of breast cancer, non-small cell lung cancer and melanoma [3,4], and their subsequent immunotherapy development has revolutionised management and outcomes. The Ovarian Cancer Action’s International Research Meeting specifically identified the tumour microenvironment’s antitumour response as an important future research focus [5].

The tumour microenvironment (TME) contributes to both anti-tumour responses and tumour progression. To date, T cells provide the largest body of evidence of the impact on ovarian cancer. Strong evidence has demonstrated an association between increased CD8+ cytotoxic T cells [6,7,8], CD4 T helper cells and improved survival rates [9]. Conversely, CD4 T regulatory cells are associated with reduced survival [9]. CD3 T cell co-receptors have been considered less frequently, demonstrating an increase in survival rates with moderately strong data [2,8]. A volume of evidence supporting the role of B cells is comparatively lacking; early evidence implies a reduced survival in association with naive B cells [10], while mature B cells may improve survival [11], although results are not unanimous. CD3 and CD20 were thus chosen as a marker for assessment to add to the existing literature. Macrophages have a disproportionately low reflection in the literature considering their abundance in the TME, where they make up the highest proportion of immune cells. The prognostic impact of tumour-associated macrophages (TAMs) in ovarian cancer is contradictory. Some studies demonstrate improved prognosis related to high TAM levels [12], while others associate M2 with reduced survival [13,14]. Their role has been proven in the carcinogenesis of multiple other primary tumours [15]. Such a lack of clarity renders TAMs interesting for analysis. 

Historically, the assessment of cells within the TME has been split into intratumoral (or intraepithelial) and stromal cells. Intratumoral cells are defined as those lying within the tumour nest, penetrating the tumour ducts and having cell-to-cell contact with no intervening stroma [4]. Stromal cells lie in the non-malignant surrounding area and do not directly contact cancer cells or tumour nests [4,6]. The specific location of tumour-infiltrating lymphocytes (TILs) within the TME is of importance [8]. Other primary tumours, most notably breast cancers, have shown an obvious preference for topography. Superiority of either localisation in ovarian cancer assessment, however, has yet to be proven. 

Touching lymphocytes are defined as lymphocytes lying at the invasive margin ”touching or within one lymphocyte cell thickness from the malignant ducts’ basement membrane” [16]. Touching lymphocytes are likely to be undergoing recruitment into the tumoral nest, hence directly reflecting the host immune response. Promising results have been seen regarding the impact of touching lymphocytes in breast tumours [17,18]. Touching lymphocytes are valuable study parameters, having demonstrated easy and quick assessment, strong concordance between assessors, and accurate representation of whole specimens [17]. Furthermore, touching lymphocyte quantification is not affected by the comparative size of the stromal or tumoral compartments. To our knowledge, touching lymphocytes in the ovary have been assessed only once before, where our research team considered their role in young women. In this small subgroup, no association was found with a histological subtype or survival. 

## 2. Results

A total of 138 cases were included in the final analysis. Clinical characteristics as demonstrated in Table 1 were diverse but representative of the general population of the United Kingdom. White British ethnicity represented the highest proportion of patients (88%). The majority of tumours were high-grade (71.74%). Half the cohort experienced advanced disease (stage 3–4) at diagnosis (52.9%). Serous cancers made up the majority of the cohort (57%). A total of 83% of patients underwent primary surgery (n = 115) with 23 receiving primary chemotherapy. Average follow-up was 63.28 months.

Overall cell abundance (mean cell infiltration percentage per high power field (HPF)) at HPF4 ranged from 0 to 50%, with a mean score of 2.22%. Lymphocytes were comparatively less represented than macrophages. Percentage presence and total cell number of all cell types was higher in the stroma when compared with the tumoral compartment. CD3-positive lymphocytes were present in 97% of slides assessed; however, their infiltrative density was low (mean percentage infiltration of 13%). CD3 cells were thus sparse among the tumour microenvironment. CD20-positive lymphocytes were the least represented cell type, present in 65.9% of slides assessed, with sparse abundance (mean percentage slide infiltration of 1%). Stromal and intratumoral CD20 cell counts were highly diverse (mean stromal count range of 0–362, mean tumoral count range of 0–368). CD68 was universal among slides. CD68 was the most densely abundant cell type (infiltration percentage of 36%). CD163 was present in the vast majority of slides (99.28%). Representative photographs of the tumour microenvironment cellular infiltrates are demonstrated in Figure 1.

### 2.1. Cellular Markers According to Histological Subtype, Stage and Grade of Disease

Serous carcinoma was the most immunogenic tumour, with CD3, CD20 and CD68 all demonstrating major associations with this subtype. CD3 mean infiltration percentage (*p* = 0.02), total stromal count (*p* = 0.02), intratumoral average (*p* ≤ 0.001) and touching average (*p* ≤ 0.001) were highest in serous carcinoma (Figure 2). Similarly, CD20 mean infiltration percentage and stromal average were highest in serous carcinoma (*p* = 0.04). Furthermore, CD68 intratumoral average was highest in serous carcinoma (*p* < 0.001). Mixed carcinoma also associated with higher CD3 stromal aggregates (*p* = 0.02), CD20 mean infiltration percentage and CD20 stromal average (*p* = 0.02) (Figure 2).

ANOVA one-way testing revealed a statistically significant correlation between CD3 (*p* = 0.031), CD20 (*p* = 0.037) and CD68 (*p* = 0.040) mean percentage infiltration according to the histological subtype (Appendix A). CD163 demonstrated no correlation with differing histological subtypes.

High-grade tumours correlated with mean stromal CD3 (*p* = 0.009), mean stromal CD20 (*p* = 0.009) and CD20 infiltration percentage (*p* = 0.036) when compared with a low-grade disease. Neither intratumoral lymphocytes nor macrophages, regardless of the site, correlated with the tumour grade. Higher-stage disease correlated with higher mean stromal CD3 when compared with low-stage (1–2) disease (*p* = 0.047). No correlation was seen between the stage of disease and intratumoral CD3, or CD20 or macrophages. Thus, while stromal lymphocytes demonstrated significant associations with both the grade and stage of the disease, such correlation was not seen with intratumoral cells.

### 2.2. Comparison between TILs Expression in Borderline Ovarian Tumours versus Cancer

Malignant tumours were consistently more immunogenic than borderline tumours with significant associations across all cell types. Mean stromal and intratumoural CD3 were significantly higher in cancer compared with borderline tumours (*p* = 0.02 and p=0.01, respectively) (Figure 3). CD20 touching average (*p* = 0.05) and intratumoral average (*p* = 0.05) were higher in cancerous tissue; stromal CD20 did not associate. The CD68 total cell count (*p* = 0.04), intratumoral average (*p* = 0.01) and touching cell average (*p* = 0.006) were higher in cancer (Figure 3). The CD163 intratumoral average *p* = 0.01), aggregates (*p* = 0.01) and touching average (*p* = 0.001) were significantly higher in cancer tissues (Figure 3).

### 2.3. TIL Patterns in Patients Who Received Neoadjuvant Chemotherapy

When TILs in tumours that received neoadjuvant chemotherapy (NACT) were compared to those that did not, the only significant results were seen in CD68. CD68 infiltration percentage and CD68 touching average were lower in tumours that received NACT (*p* = 0.02). Neither lymphocyte nor CD163 demonstrated an association.

### 2.4. Survival

Average survival was 46.24 months. A total of 96% of patients achieved one-year survival, and 53.85% of these followed up for 5 years reached 5-year survival. A total of 62% of patients are still alive at the time of writing. Some 46% of patients developed cancer recurrence within the follow-up period. Patients with endometrioid cancer demonstrated the longest mean survival (55.94 months), followed by mucinous (Figure 4). Patients with mixed cell tumours had the shortest survival (24.83 months). Patients with serous tumours survived for an average of 45.92 months.

Mean stromal CD3 count negatively correlated with survival (*p* = 0.014). Tumoral CD3 and touching CD3 did not relate. When tumours were stratified into CD3 high versus low using a median value, the higher total and stromal CD3 expression had shorter survival (*p* = 0.001 for both) (Figure 5). CD20 similarly demonstrated shorter survival with a higher mean stromal CD20 (*p* = 0.014) and CD20 infiltration percentage (*p* = 0.004) (Figure 5). Stratification according to the median again demonstrated shorter survival with higher total and stromal average CD20 (*p* = 0.001 and *p* = 0.002, respectively). Neither CD68 nor CD163 counts showed any association with survival. Similarly, touching lymphocyte scores did not associate with survival.

## 3. Discussion

Considering general cellular assessment, total CD3 counts were the highest of all cell markers assessed. CD3 abundancy, however, was lower than macrophage scores, suggesting clustering of CD3 occurs more than in TAMs. CD20 was least represented, echoing previous work [19]. CD20 counts were highly diverse, suggesting varied B cell immunogenicity among our cohort. Cellular markers were present in the vast majority of slides assessed, with larger proportional demonstrations seen than in similar previous works [11,20]. The authors consider that this is due to data collection methods as opposed to tumour characteristics. Five representative areas per slide were selected for assessment; hence, most cases demonstrated cellular representation.

The majority of significant correlations were from the stromal compartment as opposed to the tumour. Stromal TILs are considered the superior marker for assessment in other primaries, following specific guidance from the international TILs working group [4,21]. The stroma has demonstrated superior reproducibility while independent of tumoral density or growth patterns, demonstrating relative inertia in comparison with the tumoral compartment [4]. The stroma plays an important role within the TME, housing tertiary lymphoid structures and facilitating both T and B cell interactions [3]. Despite perceived superiority, tissue stroma in ovarian malignancy has been comparatively less studied, likely due to early prognostic data being taken from intratumoral counts. Our findings confirm the importance of stromal assessment in future works.

All cell types were increased in cancer versus borderline tumours. Our study group previously performed this comparison in the context of younger women and our findings echo their results [16]. Similarly, higher macrophage counts associated with cancer tissues when compared with borderline, interestingly, CD68 and CD163 touching cell counts, were significantly associated with malignant tumours. To our knowledge, this is the first study assessing macrophage presence in borderline versus malignant ovarian tumours.

Our findings demonstrated a statistically significant correlation between high-grade disease and higher CD3 and CD20 stromal averages. These results echo the existing literature; however, few studies have assessed grades across all histological subtypes [11,22,23]. We achieved significant results solely from the assessment of the stromal compartment, again demonstrating its essential role in assessment. According to the literature, higher-grade tumours do correlate with higher lymphocyte numbers in general, with less regard to their specific cellular function, unlike survival which is cell-type-specific. Indeed CD8, a known anti-tumour cell marker which positively impacts survival, and T-regs, known to reduce survival, have both been found in higher numbers in advanced-grade disease [6,23,24]. Tumour grading confers the ”aggressiveness” of a malignancy; hence, we propose that higher cell counts represent increased immunogenicity of the tumour in high-grade disease.

This study demonstrated a significant correlation between higher stromal and total CD3 expression with shorter survival. Increased CD3 presence has largely been associated with improved survival in the existing literature. Such findings are unanimously drawn from tumoral CD3, however, as opposed to stromal or combined counts [2,20,25,26,27]. Where assessed, stromal CD3 specifically has been found not to associate with survival in a handful of previous works [19,28]. These studies, however, have considered advanced-stage disease near unanimously, unlike our patient cohort. While considerably less studied than its T cell counterparts, CD20 has to date been associated with improved survival [11,29]. Our study conversely showed a statistically significant negative correlation between both stromal, total and percentage infiltration of CD20 with survival, a novel finding. On review of the aforementioned studies, the majority included only advanced-stage, high-grade, serous ovarian cancers which had undergone attempted debulking surgery. Furthermore, all studies with a positive correlation between CD20 and survival included only tumoral CD20 assessment, despite CD20 expression being proportionally increased in the stroma [19]. Although novel, our findings should be considered in relation to the heterogeneity of our cohort, specifically with more than 50% of our patients being early stage (1 or 2), and 17% receiving primary chemotherapy. Such patients may demonstrate a different tumour microenvironment. Given the disparity in evidence, the impact of CD3 presence upon prognosis in early-stage disease cannot currently be clarified and we echo the call of Macpherson for further work in this area [7].

Serous tumours were found to be the most immunogenic histological subtype among our cohort. Our findings echo the majority of previous works [11,16,28]. While previous studies have tended to focus on either the stromal or tumoral compartments, our findings showed a significant association between rising CD3 and serous tumours in both compartments. The statistically significant association between touching lymphocytes and serous tumours is a novel finding, along with the association between higher CD20 counts and the mixed histological subtype. As is the case with the majority of previous works, case numbers of non-serous histological subtypes were proportionately lower, reducing the reliability of our data; our results do, however, echo available studies.

The current literature regarding immune response following neoadjuvant chemotherapy is contradictory [21,22,30,31,32,33,34]. Our results demonstrate significant association only with CD68 infiltration percentage per HPF and CD68 touching average, which were significantly lower in tumours of post-neoadjuvant chemotherapy patients. Information on the impact of chemotherapy on macrophages is sparse in comparison to lymphocytes and evidence regarding ovarian cancer immune response to neoadjuvant chemotherapy remains primitive. We recommend further assessment of individual cell types to expand knowledge.

Correlation of cell abundancy according to age has been performed very minimally in the literature. One study was identified which specified two cohorts according to age (< or ≥52 years) [35]. A significant increase in CD8 counts was seen in the older subgroup, with neither CD3 nor CD4 showing an association [35]. Wouters et al. considered the effect of age on CD8 + CD27 infiltration and found no association [36]. O’Neill et al. performed their study in a young population (<50 yrs) where no association was seen between CD3 + CD20 versus survival; however, specific stratification according to age was not performed [16]. Our finding of increasing stromal CD3 according to age is novel within a sparsely populated field of literature. Such an association has been hypothesised to relate to patients’ hormonal status; however, further work is required in the field for further analysis [35].

By minimising exclusion criteria only to the absolute necessities that would inhibit data collection, our data collection methods ensured a true likelihood of the UK population. A total of 51% of our cohort suffered from advanced-stage disease at diagnosis, closely mimicking the national population of the UK at 50.4% [37]. Age and grade demographics are also considered representative [36]. Inclusion of borderline tumours, albeit in relatively small numbers (n = 8), was a novel addition compared to previous works. Survival rates were somewhat higher in comparison with the general population (1 year survival 96.38% vs. 71.7%, 5 year survival 53.85% vs. 46%) [38]. A total of 17% of the study patients underwent primary chemotherapy (n = 23), a high proportion in comparison to most study populations. Following Zhang’s initial reporting, optimally debulked patients have been demonstrated to show a more distinct survival advantage when associated with TIL presence [2]. Numerous studies therefore include only patients undergoing primary cytoreduction surgery. Such patients are likely to demonstrate an improved survival given both their suitability to undergo primary cytoreduction, and the resultant survival advantages. Given the aforementioned focus on a representative patient population, we consider the results appropriate for those likely to require targeted immunotherapies in the future.

The study’s rigorous data collection methods are considered a strength. Despite the rapidly evolving data available in the field, studies remain unstandardised. Not only does the location of studied TILs differ widely, so too does the reporting of cellular presence. Many studies to date employ categorical data as a scoring system to estimate the size of the cell infiltrate, or categorise cell counts at the statistical analysis stage. Where categorical data are used, cut-off values also vary highly. By profiling exact cell counts, reduction in accuracy is negated. Utilisation of both haemotoxin- and eosin stained slides and immunohistochemical staining allowed an accurate definition of the stromal and tumoral compartments, and clear cellular identification, while utilising the relevant quality-assured protocols used routinely in our laboratory for diagnostic work.

We recognise the limitations of this novel dataset. Despite selection of the most representative tumour blocks, heterogeneity in the volume of stroma vs. tumour is present, resulting in differing proportions from which to score. Although a representative array of histological subtypes was achieved, individuals with less common subtypes were less represented. Reassuringly, however, our results echo those of previous studies, improving their value when considered among the wider body of evidence available. Similarly, the number of patients undergoing adjuvant chemotherapy is small, challenging the representativeness of these results. Future work expanding the scale of the dataset with a multicentre study is encouraged, facilitated by the use of computer analysis.

## 4. Materials and Methods

All patients (n = 233) diagnosed with ovarian cancer over a 60-month time period at the Royal Wolverhampton NHS Trust, a large UK cancer tertiary referral institution, were identified. Included were epithelial ovarian primary malignant and borderline ovarian tumours with representative tumour tissue available. Inclusion and exclusion criteria are outlined in Table 2. A total of 138 patients were included in the final analysis.

### Data Collection and Laboratory Analysis

Retrospective clinicopathological data collection was performed by a single data collector from the electronic patient record. Information on grade, stage at diagnosis, age at diagnosis, ethnicity, histological subtype, neoadjuvant chemotherapy and avastin status, surgery type and date, adjuvant chemotherapy and avastin status, CA125 at diagnosis, BRCA status, comorbidities, initial treatment success, recurrence presence, recurrence interval, recurrence treatment, recurrence treatment success, 1-year survival, 5-year survival and total survival was collected. Survival data were recorded up until 3 August 2022. All patients underwent standard treatment according to gynaecological-oncological multidisciplinary team consensus. The study was carried out according to the rules of the Declaration of Helsinki of 1975. All experiments were approved by the West Midlands—Black Country NRES Committee (07/Q2702/24).

Tissue slides were reviewed and representative paraffin-embedded formalin-fixed tumour tissue blocks were selected (AS) under supervision from a senior specialist gynaecological pathologist (AB). Tissue collected at the surgical resection was prioritised over tissue biopsy (n = 3) owing to the superior volume of tissue available for analysis. Where a single block was not adequately representative, two separate blocks were collected. A representative tumour haemotoxin and eosin slide was reviewed for initial assessment of tumoral/stromal representation and recording of overall percentage cell infiltrate. Tumour sections were immunohistochemically stained for CD3 (T lymphocytes), CD20 (B lymphocytes), CD68 (pan macrophages) and CD163 (M2 macrophage subtype). Using ready to use CD3, CD20, CD68 PG-M1, CD163 antibodies, staining was done following the manufacturer’s instructions and according to the published protocols (NF). Briefly, representative 4 micron sections were cut and stained for CD3 and CD20 using a Dako Omnis. CD68 was stained using an Agilent anti-human CD68 ready-to-use antibody, clone PG-M1 (product GA613, CE-IVD), visualised with the Agilent EnVision FLEX DAB detection kit (Agilent Technologies UK Ltd). CD68 assays were performed on an Agilent OMNIS platform (Agilent Technologies UK Ltd.). CD163 was stained with a Leica anti-human CD163 concentrate antibody, clone 10D6 at a dilution of 1:200 (product NCL-L-CD163, CE-IVD) and visualised with an ultraView universal DAB detection kit. CD163 assays were performed with a Roche Ultra platform (Roche Diagnostics UK and Ireland).

Manual visual assessment was carried out by a single assessor (AS) supervised by a senior pathologist (AMS). Both the assessor and supervisor were blinded to patient details. Scoring was performed according to the International Immuno-Oncology Working Group guidelines [3], additionally differentiating between stromal, intratumoral and touching cell counts. Stromal lymphocytes were defined as cells within the tumoral borders, including the fibrovascular cores of papillary structures. Intratumoral lymphocytes were defined as intra-epithelial cells situated within the tumour nest. Touching lymphocytes were defined as those residing in the stroma either in direct contact with the tumor basement membrane, or within a one-cell diameter as previously described [17]. Aggregates were defined as a prominent cellular cluster visible at ×4 magnification. The total number of cells in the largest recorded aggregate of each slide was recorded as a hotspot. The same system was adopted for macrophage scoring. Initially, an overall infiltration percentage expression score was assessed by eyeballing stained haemotoxin and eosin sections at ×4. Scoring of immunohistochemical slides for each antibody was performed manually for each slide according to eight criteria (Table 3). Areas of necrosis, crush artefact or regressive hyalinization were avoided. Cell counts were recorded from five high-power fields (×40) as selected by the primary assessor to be representative of the slide as a whole. A mean score across the high-power fields was then calculated. Where two representative slides were collected for patients, their scores were individually collected then averaged to a single score.

Statistical analysis was performed using SPSS (IBMS statistics v 28). Chi-squared statistical analysis was used to calculate univariate analyses. The nonparametric independent samples median test (Mann–Whitney-U Wilcoxon (rank sum) test (MWW)) and Kruskal–Wallis test (KWt) were performed to compare the means and medians of the independent variables, including cell counts and clinicopathological parameters. A Spearman’s rank correlation coefficient (r_s_) was used to correlate continuous variables of biomarker expression with borderline vs. ovarian cancer. A *p* value of *p* ≤ 0.05 was used for statistical significance. Kaplan–Meier analyses were utilised to assess the overall and disease-specific survival. The data presented in this study are available on request from the corresponding author.

## 5. Conclusions

A patient cohort selected for their clinical applicability demonstrated a representative reflection of the UK population. Inclusion specifically of early-stage patients, varying histological cancer subtypes and receipt of neoadjuvant chemotherapy was novel in comparison to most previous works. Detailed data collection according to previously published international standards increased both the reliability and reproducibility of this work. Stromal assessment produced the majority of significant results across parameters assessed. Customary assessment of solely intra-tumoral cells should be challenged, with the addition of stromal scoring recommended. Survival was significantly reduced in association with high CD3 (stromal and total) and CD20 (stromal and total) and reduced survival. While prognostic results do not echo that of the bulk of the available literature, rigorous methodology, however, would suggest the reliability of our results. The differing clinicopathological characteristics included in our study must therefore be taken into consideration. Further work considering a wider patient cohort is encouraged to further consider this notion.

## Figures and Tables

**Figure 1 ijms-25-04524-f001:**
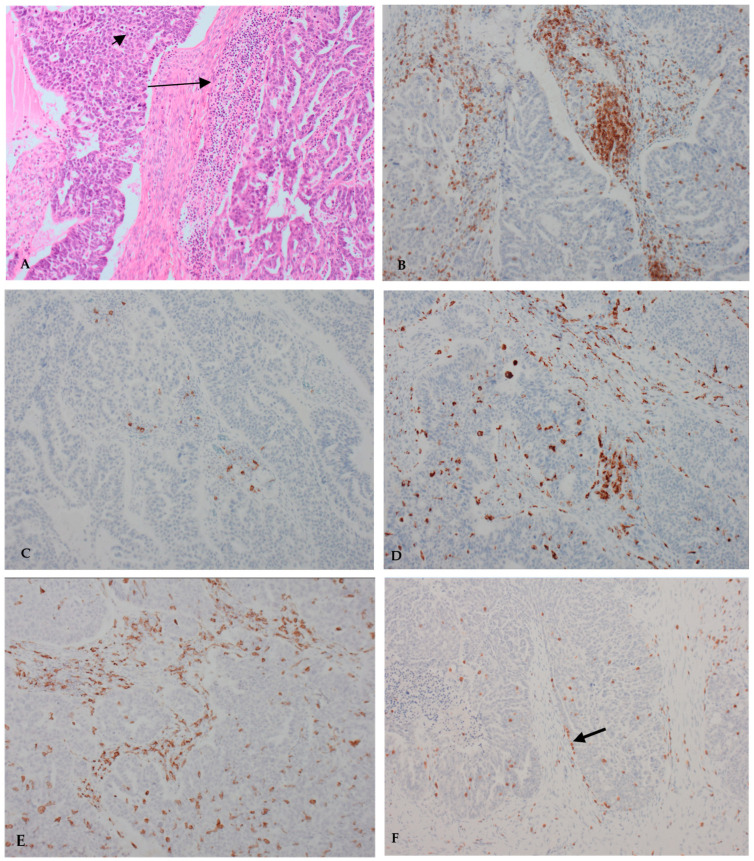
Representative photographs of cellular infiltrations for assessment at magnification ×100. Photographs A–E taken from same case. (**A**) Haemotoxin and eosin staining demonstrating stromal (long arrow) and tumoral (short arrow) tumour infiltrating lymphocytes; (**B**) Immunohistochemical staining CD3-positive stromal TILs, intratumoral TILs and a CD3-positive stromal aggregate; (**C**) Immunohistochemical staining of CD20-positive stromal and intratumoral TILs; (**D**) Immunohistochemical staining of CD68-positive stromal and intratumoral cells; (**E**) Immunohistochemical staining of CD163-positive stromal and intratumoral cells; (**F**) Touching lymphocytes lying at the invasive margin (arrow).

**Figure 2 ijms-25-04524-f002:**
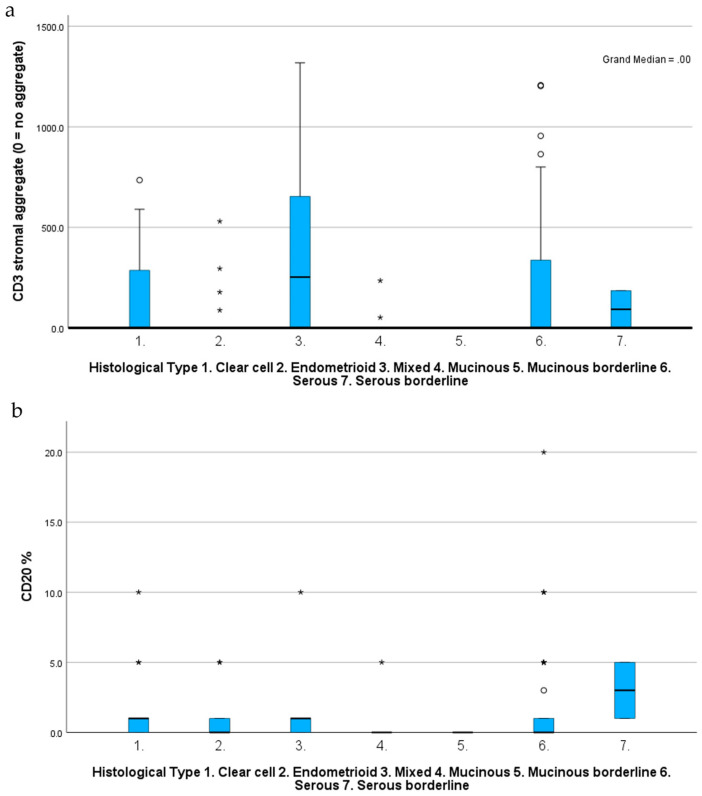
Histogram demonstrating correlation between (**a**) CD3 stromal aggregates vs. histological subtype using independent samples median test; (**b**) mean infiltration percentage of CD20 vs. histological subtype using independent samples Kruskal–Wallis test. Asterisk denotes outliers. Circles denote extreme values. Mixed and serous tumours demonstrated the highest correlation with both markers.

**Figure 3 ijms-25-04524-f003:**
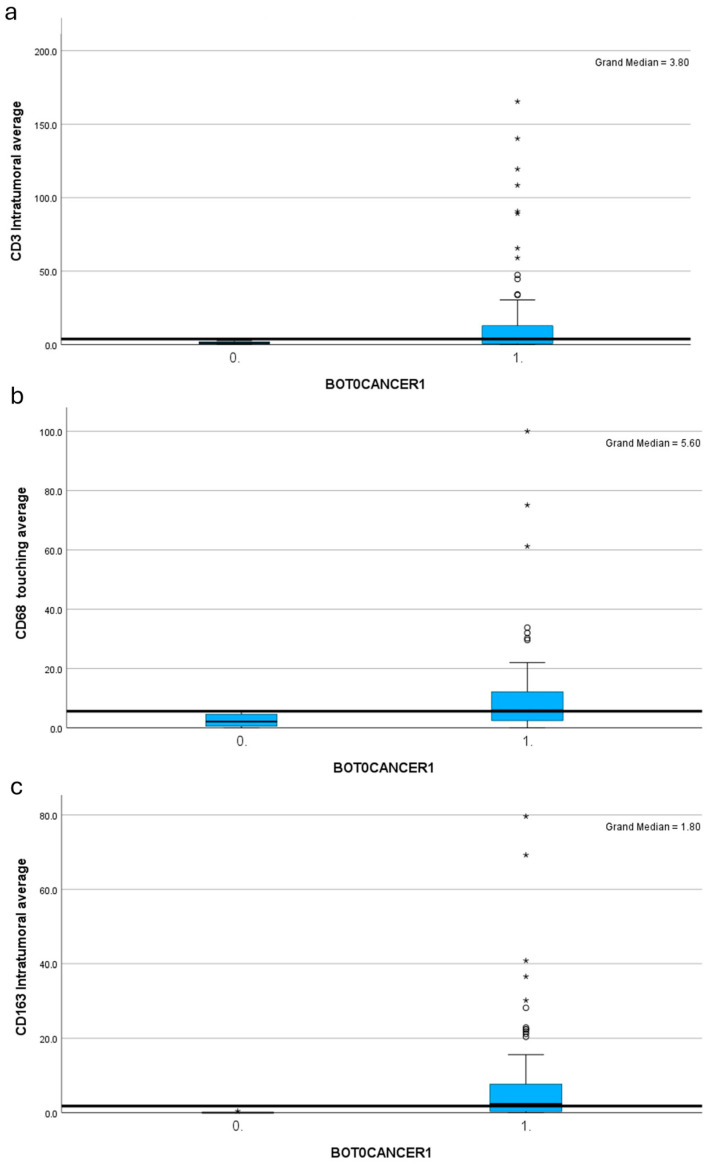
Histogram demonstrating association between mean cell counts in cancer (1) vs. borderline tumours (0) using the independent samples median test. (**a**) CD3 intratumoral average; (**b**) CD68 touching cell average; (**c**) CD163 intratumoral average. Asterisk denotes outliers. Circles denote extreme values. Significantly higher averages were seen in cancer for all markers.

**Figure 4 ijms-25-04524-f004:**
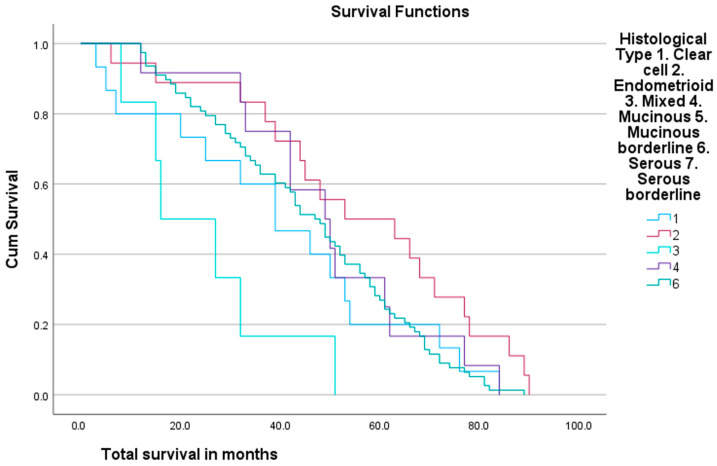
Survival according to histological subtype. Demonstrates longest mean survival in endometrial cancer patients, shortest mean survival in mixed histology cancer patients.

**Figure 5 ijms-25-04524-f005:**
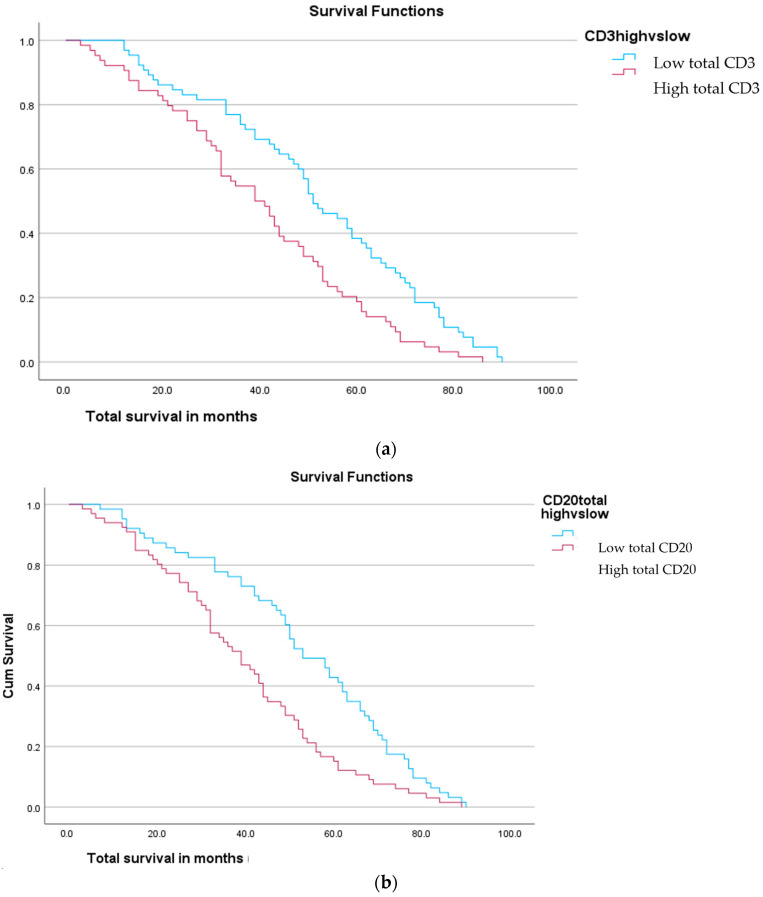
Survival curve demonstrating survival according to (**a**) CD3 total counts (high vs. low); (**b**) CD20 total count (high vs. low). Displays reduced survival with high CD3 + high CD20.

**Table 1 ijms-25-04524-t001:** Clinicopathological characteristics of the patient cohort.

**Histological Subtype**	Clear Cell	Endometrioid	Mixed	Mucinous	Mucinous Borderline	Serous	Serous Borderline
	15	18	6	12	6	79	2
**Stage**	Stage 1	Stage 2	Stage 3	Stage 4			
	53	10	60	11			
**Grade**	Grade 1	Grade 2	Grade 3	Borderline			
	19	10	100	8			
**Age**	<30 yrs	31–40	41–50	51–60	61–70	71–80	>81
	4	9	13	40	37	29	6
**Ethnicity**	White British	Indian	White Eastern European	Black Caribbean	Other Asian	Pakistani	White Southern European
	122	7	4	1	1	1	1

The mean patient age was 60.79 years (range 21–85). Patients with cancer were significantly older than those with borderline ovarian tumours (*p* = 0.007). When cell counts were assessed according to age, mean stromal CD3 correlated positively with increasing age (rs 0.172, *p =* 0.044). Tumoral and touching CD3 showed a non-significant trend towards increasing age.

**Table 2 ijms-25-04524-t002:** Inclusion and exclusion criteria at initial patient selection.

Inclusion Criteria	Exclusion Criteria
Epithelial ovarian primary tumours	Representative tumour histology not available (n = 45)
Grade 1–3 + borderline ovarian tumours	Histological subtype not epithelial (n = 14)
Stage 1–4	Primary malignancy not ovarian (n = 29)
Any or no treatment	Recurrence of previous malignancy (n = 5)
All ages	Inadequate documentation available (n = 2)

**Table 3 ijms-25-04524-t003:** Assessment and staining methods for each cell type.

Magnification	Indicator	Staining
×4	Overall percentage cell infiltrate	Haemotoxin + Eosin
×40 (mean score from five separate high-power fields)	Total cell number	Immunohistochemistry for CD3, CD20, CD68 PG-M1, CD163 antibodies
	Stromal cell count	
	Intra-tumoral cell count	
	Touching lymphocyte count	
	Stromal aggregate presence	
	Tumoral aggregate presence	
	Total cell count in largest stromal hotspot (where present)	
	Total cell count in largest tumoral hotspot (where present)	

## Data Availability

All study data are held internally by the host institution and are available upon specific request to the authors with agreement from the host institution.

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
