# Peer review of "Profiling of Tumour-Infiltrating Lymphocytes and Tumour-Associated Macrophages in Ovarian Epithelial Cancer—Relation to Tumour Characteristics and Impact on Prognosis"

_ijms, 2024, doi:10.3390/ijms25084524_

Round 1
Reviewer 1 Report
Comments and Suggestions for Authors
The authors conducted immune profiling of FFPE ovarian cancer samples using CD3, CD20, CD68, and CD163 markers. They aimed to connect immune profiles with clinical factors, histology, malignancy status, survival, and treatment response. This study has a clear, important rationale, but the presentation of results and subsequent conclusions lack the same clarity. With revisions focusing on these areas, the study's significance can be more effectively communicated.
The study's findings could greatly contribute to patient management strategies. However, the manuscript needs restructuring for clarity. Results should be presented in a more streamlined way, clearly connected to their corresponding conclusions. The authors would also benefit from revisiting figures to ensure they tell the same story as the text.
1. Figure 1F requires a more detailed legend to clearly differentiate it from 1A.
2. The study hints at the importance of CD4/CD8 T cells and T-regs, making a strong case for their inclusion in future investigations. Adding these markers could significantly enhance the conclusions.
3. The correlation between stromal CD3 counts and patient age is intriguing. Would the authors consider elaborating on potential mechanisms or hypotheses behind this in the discussion?
4. The current title is too broad given the limited immune marker panel. A more specific title should reflect the scope of the work.
The writing style is too lengthy and hard to comprehend.
Author Response
Many thanks for your considered and constructive review of our article, it was greatly appreciated. Your comments were received with interest and I have endeavoured to address all queries and requests. Below I have outlined more specifically the points raised and the subsequent alterations. To clarify the specific alterations within the manuscript I have highlighted the areas of change, including relevant line numbers in the responses below. Please note owing to the request to restructure the manuscript and alter writing style the vast majority of the paper has been re-written. We hope the changes applied enhance the quality of the manuscript.
Reviewer 1
- Focus revision to clarify results and conclusions, to more effectively communicate the study’s significance
- Results section restructured, descriptive paragraphs reworded with emphasis made to important points (p2 line 94- p8 line 205), necessary conclusions reworded (p13 line 404-406)please note these changes have been made throughout the manuscript hence individual highlighting of alterations to writing style was not performed)
- Restructure manuscript to make it clearer, alter writing style, currently too lengthy
- Introduction, results and discussion structure restructured. Writing style adjusted as requested p2 line 48-81, p2 line 94- p8 line 205, p8 line 207 – p11 line 328, p11 line 330-335, p12 line 354-365, p13 line 404-406) (please note these changes have been made throughout the manuscript hence individual highlighting of alterations to writing style was not performed). Descriptive information discussing stromal vs tumoral assessment moved from introduction to discussion (p8 line 217- p9 line 225).
- Consider going through all the data figures and trying to clarify further
- Descriptive results paragraphs reworded and reformatted. Figure legends amalgamated with further text clarification added. (p4 line 147-150; p5, line 172-175; p7 line 191-192; p8 line 204-205)
- Streamline results
- Performed as requested throughout results section (p2 line 94- p8 line 205)
- Clearly connect results to conclusion
- Performed as requested throughout results section (p2 line 94- p8 line 205, p13 line 404-406)
- Revisit figures and ensure they tell same story as text
- Figure legends amalgamated and with further text clarification added (p4 line 147-150; p5, line 172-175; p7 line 191-192; p8 line 204-205). Descriptive results paragraphs adjusted (p2 line 94- p8 line 205)
- Figure 1F requires a more detailed legend to clearly differentiate it from 1A
- Legend altered + arrow added to further clarify (p4 line 131)
- The study hints at the importance of CD4/CD8 T cells and T-regs, making a strong case for their inclusion in future investigations. Adding these markers could significantly enhance the conclusions
- Owing to resource availability we are sadly not able to add these markers
- The correlation between stromal CD3 counts and patient age is intriguing. Would the authors consider elaborating on potential mechanisms or hypotheses behind this in the discussion?
- A literature review has been performed regarding this association, summary provided as a new paragraph in the discussion (p10 line 282-292)
- The current title is too broad given the limited immune marker panel. A more specific title should reflect the scope of the work
- Title altered (p1 line 2-4)
Reviewer 2 Report
Comments and Suggestions for Authors
1. Rearrange the abstract and headings in the manuscript according to the IJMS journal writing rules.
2. Line 36: Correct definition. Mean stromal CD3, total CD3, stromal CD20 and total CD0???.
3. Although "TIL" was used as a keyword in the manuscript, it was not mentioned in the abstract. Either use this keyword in the abstract or remove it from the keywords if it is not essential. Also, use the TIL definition if it is stated in keywords.
4. Line 38: "Assessment of the stromal compartment produced a considerably higher proportion of significant results when compared to the intratumoural infiltrates." Write the sentence more clearly so that readers can understand it. In what respects were significant differences or results found?
5. Why were the markers "CD3; CD20; CD163; CD68" chosen for this study? (macrophages (CD68+), T-cells (CD3+) and B-cells (CD20+)). Please briefly state this in the abstract and detail in the introduction in a way that someone reading the article for the first time can understand.
6. In Figure 2B, the 5th and 7th data bars are missing. Is there something we are missing here?
7. Can we statistically show the correlation between CDs and histological subtypes with correlation analysis in Figure 2?
8. Please re-evaluate the manuscript for data deficiencies and spelling errors.
Author Response
Many thanks for your considered and constructive review of our article, it was greatly appreciated. Your comments were received with interest and I have endeavoured to address all queries and requests. Below I have outlined more specifically the points raised and the subsequent alterations. To clarify the specific alterations within the manuscript I have highlighted the areas of change, including relevant line numbers in the responses below. Please note owing to a request from a separate reviewer to restructure the manuscript and alter writing style the vast majority of the paper has been re-written. We hope the changes applied enhance the quality of the manuscript.
- Rearrange the abstract and headings in the manuscript according to the IJMS journal writing rules
- Performed as requested, compliant with template
- Line 36: Correct definition. Mean stromal CD3, total CD3, stromal CD20 and total CD0?
- Data information has been adjusted to clarify (p1 line 37-38)
- Although "TIL" was used as a keyword in the manuscript, it was not mentioned in the abstract
- Abstract (P1 line 28) and introduction (P2 line 79-80) altered accordingly
- Line 38: "Assessment of the stromal compartment produced a considerably higher proportion of significant results when compared to the intratumoural infiltrates." Write the sentence more clearly so that readers can understand it. In what respects were significant differences or results found
- Conclusion statement reworded (p13 line 404-406), I have kept this brief given its location within the concluding paragraph, but further clarified in the results (p3 line 111-112) and discussion (p8 line 216 - p9 line 225) section the superior role of the stromal compartment
- Why were the markers "CD3; CD20; CD163; CD68" chosen for this study? (macrophages (CD68+), T-cells (CD3+) and B-cells (CD20+)). Please briefly state this in the abstract and detail in the introduction in a way that someone reading the article for the first time can understand
- Abstract (p1 line 32) and introduction (p2 line 65-68, 71-73) altered to include reasoning for marker selection
- In Figure 2B, the 5th and 7th data bars are missing. Is there something we are missing here?
- Many thanks, new figure with all required data has replaced original (figure 2b, p9 line 146)
- Can we statistically show the correlation between CDs and histological subtypes with correlation analysis in Figure 2?
- Statistical analysis of correlation demonstrated in supplementary tables 1+2 (new addition) with explanatory sentence in text body (p5 line 142-144)
- Please re-evaluate the manuscript for data deficiencies and spelling errors
- Performed as requested
I look forward to receiving your response to our updated article and hope it meets your approval.